# *Aedes aegypti* Beta-1,3-Glucan-Binding Protein Inhibits Dengue and ZIKA Virus Replication

**DOI:** 10.3390/biomedicines12010088

**Published:** 2024-01-01

**Authors:** Xiaoxue Xie, Di Wang, Bo Li, Guorui Liang, Xiaoli Chen, Dan Xing, Teng Zhao, Xinyu Zhou, Chunxiao Li

**Affiliations:** 1State Key Laboratory of Pathogen and Biosecurity, Beijing Institute of Microbiology and Epidemiology, Beijing 100071, China; xiexiaoxueld@163.com (X.X.); m17806280983@163.com (D.W.); libo960218@hotmail.com (B.L.); lgr980210@163.com (G.L.); 15186723746@163.com (X.C.); xingdan93@163.com (D.X.); zhaoteng2013@163.com (T.Z.); 2School of Public Health, The Key Laboratory of Environmental Pollution Monitoring and Disease Control, Ministry of Education, Guizhou Medical University, Guiyang 550025, China

**Keywords:** β-1,3-glucan-binding protein, CRISPR/Cas9, overexpression, DENV2, ZIKV

## Abstract

GNBPB6, a beta-1,3-glucan-binding protein, was identified in the transcriptome of *Aedes aegypti* (*A. aegypti*) with dengue (DENV), Zika (ZIKV), and chikungunya viruses (CHIKV). In this study, we not only clarified that DENV2 and ZIKV regulate the changes in *GNBPB6* expression but also identified the relationship of this gene with viral infections. The changes in *GNBPB6* expression were quantified and showed a decrease in *A. aegypti* cells (Aag2 cells) at 2 dpi and 3 dpi and an increase at 4 dpi and 5 dpi (*p* < 0.05). A significant increase was observed only at 5 dpi after DENV2 infection. Subsequently, a *GNBPB6* knockout (KO) cell line was constructed using the CRISPR/Cas9 system, and the DENV2 and ZIKV RNA copies, along with cell densities, were quantified and compared between the KO and wild type (WT) cells at different dpi. The result showed that DENV2 and ZIKV RNA copies were significantly increased in the KO cell line with no significant change in cell growth. Finally, DENV2 copies decreased after *GNBPB6* was complemented in the KO. In conclusion, *GNBPB6* knockout and complementation in Aag2 cells revealed that *GNBPB6* can inhibit the replication of both DENV2 and ZIKV. These results contribute to subsequent research on mosquito–virus interactions.

## 1. Introduction

*Aedes aegypti* (*A. aegypti*) are vectors of dengue virus (DENV), Zika virus (ZIKV) and chikungunya virus (CHIKV) [1] and have given rise to significant global public health issues due to their widespread distribution [2]. DENV and ZIKV, which belong to the *Flavivirus* genus and *Flaviviridae* family of viruses, have similar infection processes [3]. With geographical range expansion by urbanization, climate change, and global travel, it is improbable that arboviral outbreaks will abate in the future [4]. Without efficient vaccines and curative drugs, focusing on this common vector is the optimal strategy for controlling the spread of both viruses [5]. However, the current vector control method of using chemical insecticides is ineffective in preventing outbreaks, partly due to insecticide resistance [6]. In recent years, with the growth of technologies for genome editing and mosquito transgenics, emerging prevention and control strategies based on population inhibition or population displacement have been proposed as alternative options for controlling mosquito-borne diseases [7,8].

*A. aegypti* beta-1,3-glucan-binding protein (AAEL007064) is also known as Gram-negative bacterial binding protein B6 (GNBPB6). Previous studies have identified changes in *GNBPB6* transcription in various tissues of *A. aegypti* mosquitoes infected with DENV2, ZIKV and CHIKV. *GNBPB6* expression was upregulated 1.85-fold (Log_2_ fold change = 0.890) in the *A. aegypti* carcasses with DENV2 infection [9]. However, the opposite trend was observed after infection with ZIKV and CHIKV, with Log_2_ fold changes of −0.715 and −1.058, respectively [10]. Although changes in *GNBPB6* have been reported in these studies, they focused more on the antiviral signaling pathway rather than the function of *GNBPB6*. Therefore, the relationship between *GNBPB6* and viral infection is currently unclear.

*A. aegypti* GNBPB6 belongs to the Gram-negative bacterial binding protein family, which is associated with mosquito innate immunity. GNBP is a major pattern recognition receptor (PRR) in insects and a wide range of invertebrates [11,12,13,14,15]. It has been demonstrated that lipopolysaccharide and β-1,3-glucan can stimulate increased GNBP expression, which then activates the prophenoloxidase-activated system (proPO-AS) [16]. This activation is associated with increased oxidative activity, antimicrobial activity, as well as signaling pathways against Gram-positive bacteria [17,18] and fungi [19,20,21]. The mosquito’s GNBP has been found to be related to Plasmodium [22] and arbovirus infection [23] in *Anopheles gambiae* [24], *A. aegypti* [25], and *Culex pipiens quinquefasciatus* [26]. *GNBPB3* and *GNBPB4* function as anti-DENV in *A. aegypti* by activating the Toll signaling pathway [27]. Homology to beta-1,3-glucan binding proteins has also been studied in other species. For example, lipopolysaccharide and beta-1,3-glucan binding proteins bind to heat shock protein 70 and activate proPO-AS in shrimp [28], while plasma beta-1,3-glucan binding proteins mediate opsono-phagocytosis in marine mussel *Perna viridis* [29]. These studies suggest that beta-1,3-glucan binding proteins have multiple functions.

These studies suggested that some members of the *A. aegypti* GNBP family are associated with innate immunity. However, the available studies only demonstrated that *GNBPB6* may be involved in the regulation of arbovirus infection at the transcriptome level, and the specific function of *GNBPB6* is unclear. Therefore, in this study, the *GNBPB6* knockout (KO) Aag2 cell line was constructed using the CRISPR/Cas9 system, and changes in viral RNA copies were detected in the KO cell line and compared to those in wild-type (WT) cells. Through these analyses, the function of *GNBPB6* in controlling DENV2 and ZIKV replication in Aag2 cells was clarified.

## 2. Materials and Methods

### 2.1. Virus, Cells and Plasmids

The DENV2 Guangdong strain was provided by Guangdong Provincial Centers for Disease Control and Prevention [30]. Zika virus strain SZ01 (GenBank: KU866423) is stored in the microbial culture collection center of the Institute of Microbiology and Epidemiology in Beijing.

*A. aegypti* Aag2 cells were cultured in Schneider’s Drosophila Medium (SDM, Gibco, Thermo Fisher Scientific, Waltham, MA, USA) and supplemented with 10% fetal bovine serum (FBS, Gibco, Thermo Fisher Scientific, Waltham, MA, USA) at 28 °C with 5% CO_2_.

PSL1180polyUBdsRED was a gift from Leslie Vosshall (Addgene plasmid # 49327; http://n2t.net/addgene:49327; RRID: Addgene_49327, accessed on 1 February 2021.) [31]. Snapgene 4.1.8 software was used to design experimental and control plasmids using the double enzyme digestion method to avoid random cutting and reverse ligation. EGFP and puromycin sites were added to the original vector, and EGFP was used as a control for the experimental overexpression vector experiments (Appendix A). The reporter gene, DsRed, was used to show the expression and location of the target gene in cells. In the experimental plasmid vector, the EGFP sequence was replaced with the protein coding region of the gene of interest (Appendix A). After the vector was designed, it was synthesized by Sangon Bio (Shanghai, China).

### 2.2. GNBPB6 Expression Changes in Different Tissues of A. aegypti in Response to DENV, ZIKV, and CHIKV

GNBPB6 expression was increased in the transcriptome of *A. aegypti* SGs infected with DENV in our previous study (unpublished). It was also identified in the midgut and SGs transcriptomes of *A. aegypti* infected with DENV, ZIKV, and CHIKV (Table 1). The changes in *GNBPB6* mRNA expression levels after virus infection were quantified by fold change (FC) in comparison with the uninfected condition.

### 2.3. Ribonucleoprotein Transfection

The DNA sequence of *A. aegypti GNBPB6* was obtained from the NCBI database, and small guide RNA (sgRNA) sequences were designed and synthesized by GenScript Biotech. The sgRNA1 target sequence was 5′-GCCAAACATTTCTGTCCAGT-3′, whereas the sgRNA2 target sequence was 5′-ATTGGTTGTGGCCTGCTCTG-3′. Chemically synthesized sgRNA and Cas9 protein (Invitrogen, Thermo Fisher Scientific, Waltham, MA, USA) were combined to form in vitro ribonucleoprotein (RNP) complexes, which were delivered into Aag2 cells by electroporation. Specific methods are described in Soyoung A. Oh et al. [32]. The collected Aag2 cells were resuspended in a mixture of RNP complex and nucleofection buffer (Lonza, Basel, Switzerland) and electroporated in the K562 program of Nucleofector (Lonza, Basel, Switzerland) for five minutes. Immediately, prewarmed SDM containing 10% FBS was added to the nucleofection cells in Petri dishes and mixed gently by pipetting once or twice. Individual cell clones were subsequently selected under a microscope and placed in 96-well plates.

### 2.4. DNA Extraction and GNBPB6 DNA Amplification

After the cells had grown to a specific quantity, the Aag2 cell DNA was extracted using the insect DNA extraction kit (Enlighten Biotech, Shanghai, China) and amplified by PCR with the designed primer sequences (forward primer: 5′-AGGTGCCAATTGACCAAACA-3′ and reverse primer: 5′-AACTGAATCCGCCAAGTCTC-3′), and then the amplified products were subjected to agarose gel electrophoresis and sequencing. The PCR reaction included the following: 2× EasyTaq^®^ PCR SuperMix (TransGen Biotech, Nanjing, China) (12.5 μL); 10 μM forward primer (0.5 μL); 10 μM reverse primer (0.5 μL); DNA template (2 μL); and nuclease-free water to a total volume of 25 μL. The reaction procedure was as follows: 94 °C for 10 min, then 35 cycles at 94 °C for 30 s, 60 °C for 30 s, 72 °C for 1 min 15 s, and 72 °C for 10 min. The PCR products were subjected to agarose gel electrophoresis and submitted to Tianyi Huiyuan Company (Beijing, China) for sequencing.

### 2.5. Aag2 Cells Infected with DENV2 and ZIKV

Five milliliters of Aag2 cell suspension with a density of 1 × 10^6^ cells/mL was inoculated into 25-cm^2^ cell culture bottles. After incubation for twelve hours, DENV2 was diluted in SDM containing 5% FBS to obtain a multiplicity of infection (MOI) of 0.01. Next, one milliliter of the diluted virus was added, and the cells were cultured in a cell incubator with 5% CO_2_ at 28 °C. After infection, 200 μL of the cell supernatant was taken daily and stored at −80 °C until the 8th day after infection for viral RNA copy numbers and titer detection. The experiment included two groups, with four biological replicates: the KO Aag2 cells + DENV2 (DKO) group and the WT Aag2 cells + DENV2 (DWT) group. Additionally, ZIKV was diluted in SDM containing 5% FBS to obtain an MOI of 0.001. The other experimental steps were similar to those for DENV2. The supernatant was taken from cells at 2, 3, 4, 5, 6, 7, and 8 days post-infection and stored at −80 °C.

### 2.6. Estimation of Viral Copies in the Cell Supernatant

DENV2 RNA copies were detected using the DENV2 nucleic acid-free detection kit (MyLab Medical Technology, Beijing, China). The reaction system included 2× DENV2 amplification solution (10 μL), DENV2 primer and probe mixture (1 μL), RT–PCR enzyme mixture (1 μL), cell supernatant (4 μL), and nuclease-free water to 20 μL. The reaction conditions were as follows: 50 °C for 10 min, 95 °C for 10 min, and 40 cycles of 95 °C for 15 s and 60 °C for 30 s. DENV2 primer sequences and the probe sequence were as follows: forward primer (5′-AATTAGAGAGCAGATCTCTGATGAA-3′), reverse primer (5′-AGCATTCCAAGTGAGAATCTCTTTGT-3′), and probe (5′-FAM-AGCATTCCAAGTGAGAATCTCTTTGTCA-BHQ1-3′). ZIKV RNA copies were detected using a ZIKV nucleic acid-free detection kit (MyLab Medical Technology, Beijing, China). The ZIKV primer sequences and the probe sequence were as follows: forward primer (5′-CGCTGCCCAACACAAGGT-3′), reverse primer (5′-CCACTAACGTTCTTTTGCAGACA-3′), and ZIKV probe (5′-FAM-AGCCTACCTTGACAAGCAGTCAGA-BHQ1-3′). The reaction system and conditions were the same as those described above for DENV2 amplification.

### 2.7. Plasmid Transfection

First, to subculture the Aag2 cells, 1 mL of cells (1 × 10^5^ cells/mL) were seeded in a 12-well plate (Thermo Fisher Scientific) and incubated in SDM containing 2% FBS for 3 days, after which transfection experiments were performed. FuGENE 6 transfection reagent (Promega, Milan, Italy) was used for transfection. The ratio of plasmid to transfection reagent was 1 µg plasmid: 8 µL transfection reagent. The transfection buffer consisted of Opti-MEM medium in addition to the mixture, and 50 µL transfection buffer was added to each well. After 24 h, red fluorescence expression was observed under a fluorescence microscope.

### 2.8. Detection of Relative Gene Expression Levels by qRT–PCR

TransScript^®^ Green One-Step qRT–PCR SuperMix (TransGen Biotech, Nanjing, China) was used for qRT–PCR detection with RNA as the template. The GNBPB6 qRT–PCR primer sequences were forward primer (5′-TGGCAGCATGAAAATTCGCT-3′) and reverse primer (5′-TGTCAATGTTGGGCGGATGT-3′). The reaction system included 2× PerfectStart^TM^ Green One-Step qPCR SuperMix (10 μL); TransScript^®^ Green One-Step RT/RI Enzyme Mix (0.4 μL); 10 μM forward primer (0.4 μL); 10 μM reverse primer (0.4 μL); 50× passive reference dye (0.4 μL); RNA template (2 μL); and nuclease-free water to a volume of 20 μL. The reaction procedure was as follows: 45 °C for 5 min, and 40 cycles of 94 °C for 30 s, 94 °C for 5 s and 60 °C for 30 s. Ribosomal protein S6 (RPS6) was used as the reference gene, and the relative expression level was calculated using the 2^−ΔΔCT^ method [33]. The reference gene primer sequences were forward primer (5′-CGTCGTCAGGAACGTATCC-3′) and reverse primer (5′-TTCTTGGCAGCCTTAGCAG-3′).

### 2.9. Statistical Methods

All the data are represented as the mean ± standard deviation (SD) or standard error of the mean (S.E.M.). Excel 2019 was used to process the data, GraphPad Prism 8.0 software was used for mapping, and IBM SPSS Statistics 25 software was used to evaluate statistical significance. For independent samples *t* tests were used to detect differences in relative gene expression levels, cell density, and viral RNA copies.

## 3. Results

### 3.1. GNBPB6 Expression Changes in Aag2 Cells with Virus Infection

Previous studies have shown varying levels of GNBPB6 expression in different tissues of *A. aegypti* infected with DENV2 and ZIKV (Table 1). In this study, GNBPB6 mRNA expression in Aag2 cells was determined at different days post-infection (dpi) (Figure 1). The fold change (FC) was used to characterize the GNBPB6 mRNA expression level relative to that of RPS6, and the level in the uninfected group was used as a control. The FC showed an immediate decline after virus infection and a subsequent increase, peaking at 5 dpi and then a gradual recovery. After ZIKV infection, GNBPB6 gene expression decreased significantly at 2 dpi (FC = 0.253, *p* = 0.013) and 3 dpi (FC = 0.359, *p* = 0.004) and increased significantly at 4 dpi (FC = 1.763, *p* = 0.034) and 5 dpi (FC = 15.986, *p* = 0.028). A statistically significant difference in FC was observed only at 5 dpi after DENV2 infection (FC = 5.347, *p* = 0.001). 

### 3.2. CRISPR/Cas9-Mediated GNBPB6 Knockout in Aag2 Cells

Two pairs of *GNBPB6* sgRNAs were designed to target the sequence between the second and fourth exons using the CRISPR/Cas9 system, and ribonucleoprotein transfection (RNP) was used to construct KO cell lines. *GNBPB6* DNA amplification and sequencing indicated a total deletion of 500 bp between the two sgRNA target sites, resulting in a code-shifting mutation (Figure 2A). The results of gel electrophoresis and DNA sequence comparisons between KO and WT cells showed that the monoclonal DNA was selected at the DNA level (Figure 2B). The FC fell to 0.0006 ± 0.0001 in KO cells (Figure 2C). Therefore, the knockout cells were confirmed to be successfully deficient in *GNBPB6*.

### 3.3. Changes in Aag2 Cell Density

The cell density of KO and WT cells was tested in four replicates. The analysis was performed in uninfected and infected groups (infected with DENV2 and ZIKV). There was no significant difference in cell density before and after GNBPB6 KO, and the density of KO cells began to decrease relative to WT cells after 5 days of cell growth (Figure 3A). There was also no significant difference in the cell density of KO and WT cells before and after virus infection (Figure 3B,C). However, the cell density of WT cells decreased after DENV2 and ZIKV infection. In addition, the density of KO cells did not differ from that of WT cells after DENV2 and ZIKV infection (Figure 3D,E).

### 3.4. GNBPB6 KO Promotes Viral Replication during Viral Infection

To investigate the function of GNBPB6 in controlling viruses transmitted by *A. aegypti*, *GNBPB6* KO and WT cells were simultaneously infected with DENV2 and ZIKV. Viral RNA copies were detected at 1, 2, 3, 4, 5, 6, 7, and 8 post-DENV2 infection in KO and WT Aag2 cells (Figure 4A). In general, viral RNA copies accumulated as the duration of infection increased. There was a significant difference in RNA copies in KO cells when compared to those in WT cells at 3, 4, and 5 dpi (*p* < 0.001) (Figure 4B). ZIKV RNA copies were detected at 2, 3, 4, 5, 6, 7, and 8 dpi in KO and WT Aag2 cells (Figure 4C). Similar results were observed in cells infected with ZIKV, where the viral copy numbers were significantly increased in the KO cells at 4, 5, 6, and 7 dpi (*p* < 0.05) (Figure 4D). These results suggested that the absence of *GNBPB6* increased viral replication in the middle stage of infection.

### 3.5. Restoration of GNBPB6 Expression in KO Cells

The constructed *GNBPB6* and control (*EGFP*) plasmids were transfected into KO cells. The cells and their fluorescence were observed and assessed (Figure 5A). While the transfection efficiency of *EGFP* was lower, significant transfection efficiency was still observed. The target gene plasmid was expressed in KO cells, and DENV2 RNA copies were detected at 2, 4, and 6 dpi. The results indicated that the DENV2 copies decreased significantly in the experimental group (expressing the plasmid of *GNBPB6*) (*p* < 0.001) compared with both the negative control (expressing the plasmid of EGFP) and the blank control (not transfected with any plasmid) groups, and there was no significant difference on day 2 and day 6 after infection (Figure 5B).

## 4. Discussion

Previous studies have suggested that GNBP family members may be involved in the challenge of bacteria in invertebrates. Several studies have also observed differential expression of some genes in the GNBP family in the immune response of insects, such as Toll [20] and immune deficiency (IMD) [34] pathways. However, it appears that these studies have focused more on the antiviral activity of these pathways rather than testing the function of the GNBPB family. In addition, the function of *A. aegypti* beta-1,3-glucan-binding protein in response to dengue virus 2 (DENV2) and Zika virus (ZIKV) has not yet been clarified. This study focuses on the inhibitory effect of a beta-1,3-glucan binding protein on the replication of DENV2 and ZIKV in Aag2 cells.

The GNBPB6 mRNA expression level was found post virus infection, with the result suggesting that GNBPB6 expression is downregulated by ZIKV infection in the early stages (at 2 and 3 dpi), upregulated in the middle stages (at 4 and 5 dpi), and then decreases to levels observed in the uninfected group during the late stages (at 6, 7, and 8 dpi). It differs from previous results downregulated by ZIKV in the SGs of *A. aegypti*. In this study, the dynamic changes of *GNBPB6* expression in response to ZIKV were observed in Aag2 cells, suggesting that the increased expression was stimulated by viral replication. A similar phenomenon was observed in DENV2 infection, with a significant increase in expression at 5 dpi, which is consistent with the results in adults. Therefore, GNBPB6 is identified as upregulated in response to DENV2 and ZIKV infection in Aag2 cells.

The function of GNBPB6 was investigated via knockout and overexpression in Aag2 cells. The GNBPB6 knockout cell line was generated using CRISPR/Cas9 technology and proved to be a valuable research model for arbovirus infections. During the middle stage of infection, the DENV2 (at 3, 4, and 5 dpi) and ZIKV (at 4, 5, 6, and 7dpi) RNA copies were significantly higher in the KO cells. At the same time, plasmids containing GNBPB6 were introduced into KO cells by an overexpression system. The DENV2 RNA copies decreased only at 4 dpi, further validating the previous conclusion.

Due to inadequate acquired immunity, mosquitoes rely predominantly on innate immunity to defend against pathogen infection [35,36]. Once pathogens cross the mosquito midgut barrier and enter the bloodstream, the recognition and binding of the pathogen-associated molecular pattern (PAMP) on the the pathogen surface and the mosquito pattern recognition receptor (PRR) activate cellular and humoral immunity [37,38], which then activates a downstream immune response to protect the mosquito from the pathogen [39]. The current study suggests that mosquito GNBPs act as PRRs that activate Toll and IMD signaling pathways. These pathways were shown to be associated with resistance to DENV2 infection in *A. aegypti*. Therefore, this study proposes a hypothesis that *GNBPB6* may act as a PRR to bind with the PAMPs on the virus surface, activating the mosquito innate immunity to exert antiviral effects. The deficiency of GNBPB6 leads to decreased innate immunity in mosquitoes, leading to a phenotype characterized by an increase in viral RNA copies within KO cells.

This study demonstrates the specific function of a single member of the *A. aegypti* GNBP family in DENV2 and ZIKV infections, and it is helpful for further understanding of arbovirus–mosquito interactions. However, a limitation also exists. The viral infection regulating mechanism of GNBPB6 is still unclear and requires further study.

## 5. Conclusions

In conclusion, this study determined that the *GNBPB6* expression changes in Aag2 cells infected with DENV2 and ZIKV and clarified the function of GNBPB6 in response to these viruses using the CRISPR/Cas9 and overexpress systems. Characterizing the function of GNBPB6 can assist in screening candidate genes for new control technologies aimed at preventing and controlling viruses transmitted in mosquitoes.

## Figures and Tables

**Figure 1 biomedicines-12-00088-f001:**
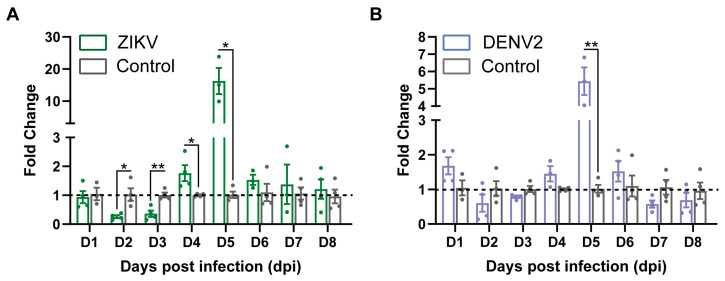
*GNBPB6* expression changes in Aag2 cells after virus infection. *GNBPB6* relative to the *RPS6* expression level was characterized by fold change (FC) compared to the uninfected group. Each dot represents a sample data. All the data are presented as the means ± SDs. Error bars represent the SDs of at least three biological replicates. Independent samples *t* tests: * *p* < 0.05, ** *p* < 0.01. (**A**) FC in wild type (WT) Aag2 cells infected with ZIKV. FC at 2, 4, and 5 dpi *p* < 0.05, 3 dpi *p* < 0.01. (**B**) FC in WT Aag2 cells infected with DENV2. FC increased at 5 dpi (*p* < 0.01).

**Figure 2 biomedicines-12-00088-f002:**
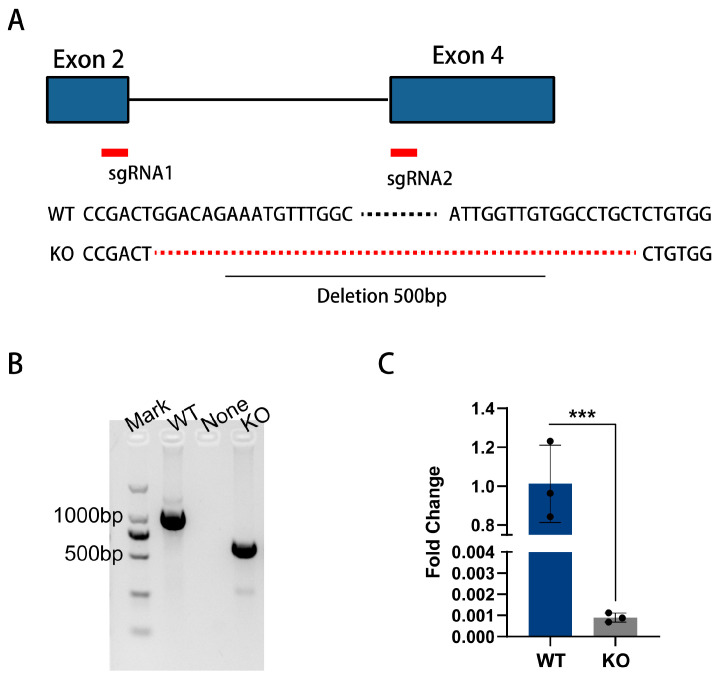
CRISPR/Cas9-mediated *GNBPB6* knockout in Aag2 cells. (**A**) Partial DNA sequences are displayed at the deletion position. Two pairs of sgRNA target sequences are observed in WT cells, and a total fragment length of 500 bp is deleted between the sgRNA pairs in the KO cells. The deleted sequence is located between exons two and four of the GNBPB6 gene DNA sequence. (**B**) Agarose gel electrophoresis results. “Mark” indicates the 2000 bp marker, “WT” is a control, length 1146 bp, and “None” is a negative control, “KO” is knockout cells, length 646 bp. (**C**) Detection of GNBPB6 mRNA expression levels in KO cells compared with WT cells. Each dot represents a sample data. All the data are represented as the means ± SDs. Error bars represent the SDs of three biological replicates. Independent samples *t* tests: *** *p* < 0.001.

**Figure 3 biomedicines-12-00088-f003:**
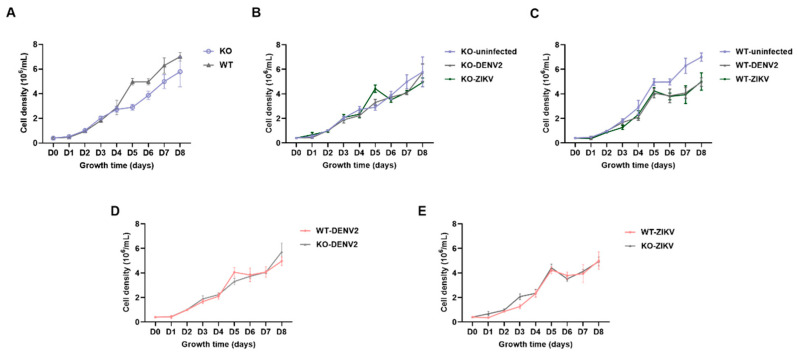
Changes in the knockout (KO) and wild type (WT) cell density before and after DENV2 and ZIKV infection. (**A**) The cell density curve of KO and WT cells. (**B**) The cell density curve of KO cells infected with DENV2 and ZIKV. (**C**) The cell density curve of WT cells infected with DENV2 and ZIKV. (**D**) The cell density curve of KO and WT cells infected with DENV2. (**E**) The cell density curve of KO and WT cells infected with ZIKV. All the data are represented as the means ± SDs. Error bars represent the SDs of four biological replicates. Independent samples *t* tests. There were no differences in cell density comparisons at each time point.

**Figure 4 biomedicines-12-00088-f004:**
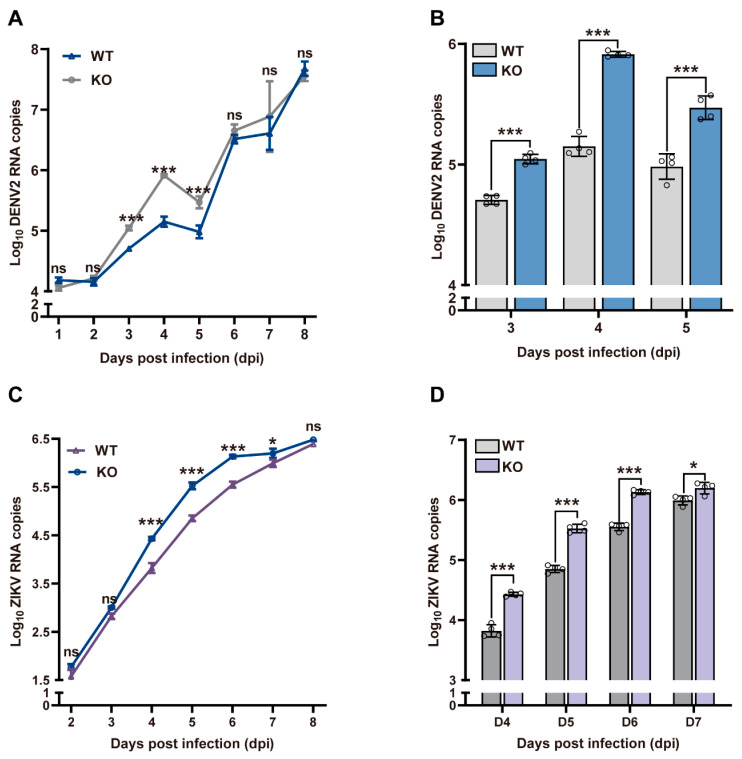
Viral replication increased in the middle stages of DENV2 and ZIKV infection. The curve of DENV2 RNA copies (**A**) and ZIKV RNA copies (**C**). Significant differences in DENV2 (**B**) and ZIKV (**D**) RNA copies. Each dot represents a sample data. All the data are represented as the means ± SDs. Error bars represent the SDs of four biological replicates. Independent samples *t* tests: * *p* < 0.05, *** *p* < 0.001, ns means no significance.

**Figure 5 biomedicines-12-00088-f005:**
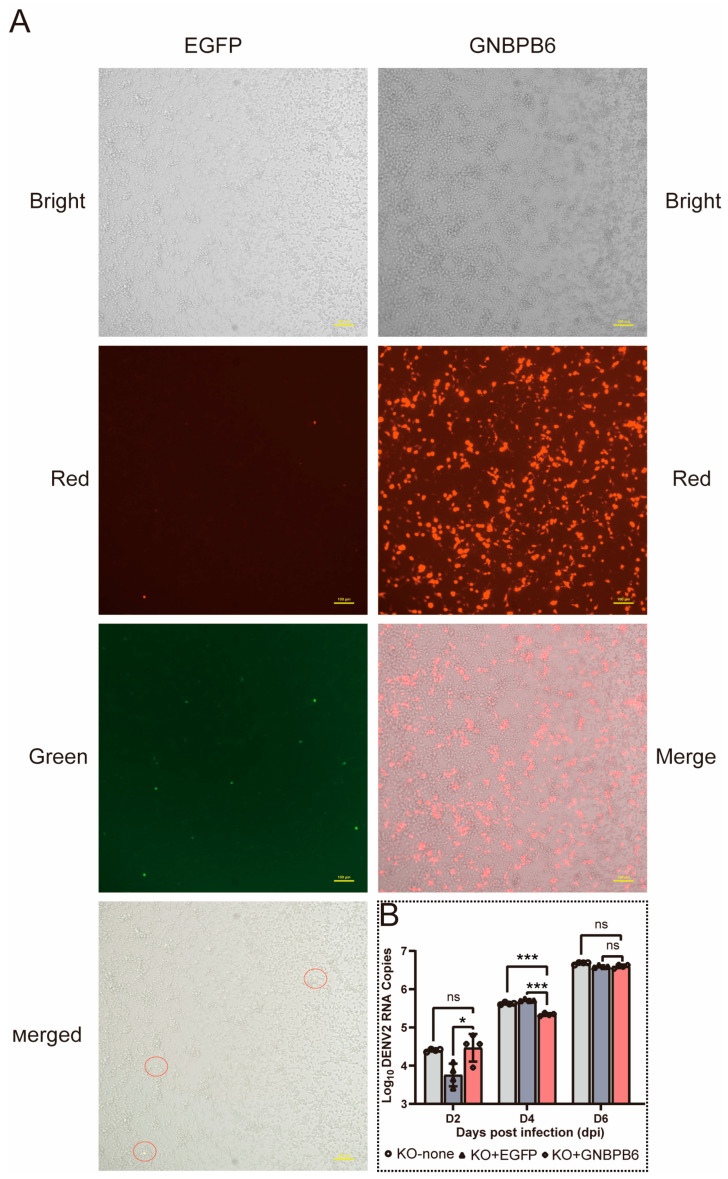
Expression of GNBPB6 in knockout (KO) cells. (**A**) Observation of fluorescence of different plasmids expressed in KO cells. EGFP displays green and red fluorescence, GNBPB6 expresses red fluorescence. Red circles are expressing both EGFP and dsRed. (**B**) Changes in DENV2 RNA copies were observed in KO cells after GNBPB6 expression; there was a significant decrease after *GNBPB6* expression (*p* < 0.001). Each dot represents a sample data. All the data are represented as the means ± SDs. Error bars represent the SDs of four biological replicates. Independent samples *t* tests: * *p* < 0.05, *** *p* < 0.001; ns means no significance.

**Table 1 biomedicines-12-00088-t001:** GNBPB6 expression changes in different tissues related to DENV, ZIKV, and CHIKV.

Virus	Tissues	Log2Fold Change	*p* Value	Reference
DENV	Carcass	0.890	<0.05	Zhiyong Xi et al. [9]
ZIKV	SGs	−0.751	<0.05	Avisha Chowdhury et al. [10]
CHIKV	SGs	−1.058	<0.05	Avisha Chowdhury et al. [10]

## Data Availability

All data generated or analyzed during this study are included in this published article.

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
