# Peer review of "Aedes aegypti Beta-1,3-Glucan-Binding Protein Inhibits Dengue and ZIKA Virus Replication"

_biomedicines, 2024, doi:10.3390/biomedicines12010088_

Round 1

Reviewer 1 Report

Comments and Suggestions for Authors

1 The entire abstract is quite confusing and does not give the right information about the rest of the manuscript

2 was identified by transcriptome 41 analysis of salivary glands (SGs) from Ae. aegypti with DENV type two (DENV2) infection....Salivary gland from who or what?

3 the overall paper seems very weak, without a stronger analysis and evaluation the risk is that the manuscript just records of in vitro procedure.

4 There are no real novelties as other authors have had similar results almost 15 years ago, Xi Z, Ramirez JL, Dimopoulos G. The Aedes aegypti toll pathway controls dengue virus infection. PLoS Pathog. 2008 

Comments on the Quality of English Language

English needs to be revisited

Reviewer 2 Report

Comments and Suggestions for Authors

Dear authors,

Your manuscript is interesting and gives new information about Zika and dengue viruses interactions with host organisms.

However, in current form the manuscript cannot be published and need many corrections:

First, some English corrections are needed throughout the text.

Second, a lot of mistakes were found:

Title of the manuscript: “Aedes aegypti beta-1,3-glucan-binding protein inhabits dengue and Zika virus replication” – what do you mean under “inhabits”? Lane 276 – the same. As far as I understand, “inhabits” usually used as meaning “live in something”?

References in text and tables should be corrected according to publisher’s requirements.

Lane 28 – “chikungunya virus” – should start with capital letter - “Chikungunya virus”

Lanes 86, 182 – “Ae. aegypti” should be in italic

Lane 86-87 – “GNBPB6 expression was increased in the transcriptome of Ae. aegypti SGs infected 86 with DENV in our previous study (unpublished)” – I’m not sure that including unpublished data is a good idea; in this case, should the reader take your word for it? Please, avoid this.

Table 1 – “(unpublic)” - this word is inappropriate here. Correct, please.

Lane 104 – “in vitro” should be in italic.

Agarose gel results – fragment lengths should be specified.

Comments on the Quality of English Language

Some English corrections are needed throughout the text
